# Proteomic Profile of Flaxseed (*Linum usitatissimum* L.) Products as Influenced by Protein Concentration Method and Cultivar

**DOI:** 10.3390/foods13091288

**Published:** 2024-04-23

**Authors:** Markéta Jarošová, Pavel Roudnický, Jan Bárta, Zbyněk Zdráhal, Veronika Bártová, Adéla Stupková, František Lorenc, Marie Bjelková, Jan Kyselka, Eva Jarošová, Jan Bedrníček, Andrea Bohatá

**Affiliations:** 1Department of Plant Production, Faculty of Agriculture and Technology, University of South Bohemia, Na Sádkách 1780, 370 05 České Budějovice, Czech Republic; jarosovam@fzt.jcu.cz (M.J.); vbartova@fzt.jcu.cz (V.B.); stupkova@fzt.jcu.cz (A.S.); jarose02@fzt.jcu.cz (E.J.); bohata@fzt.jcu.cz (A.B.); 2Mendel Centre of Plant Genomics and Proteomics, Central European Institute of Technology, Masaryk University, Kamenice 753/5, 625 00 Brno, Czech Republic; pavel.roudnicky@ceitec.muni.cz (P.R.); zbynek.zdrahal@ceitec.muni.cz (Z.Z.); 3Department of Food Biotechnology and Agricultural Products Quality, Faculty of Agriculture and Technology, University of South Bohemia, Studentská 1668, 370 05 České Budějovice, Czech Republic; lorencf@fzt.jcu.cz (F.L.); bedrnicek@fzt.jcu.cz (J.B.); 4Department of Legumes and Technical Crops, Agritec Plant Research Ltd., Zemědělská 2520/16, 787 01 Šumperk, Czech Republic; bjelkova@agritec.cz; 5Department of Dairy, Fat and Cosmetics, University of Chemistry and Technology, Technická 5, 166 28 Prague, Czech Republic; jan.kyselka@vscht.cz

**Keywords:** flaxseed, cultivar, flour, protein concentrates, proteomic profile, 11S globulin

## Abstract

The research is focused on the quantitative evaluation of the flaxseed (*Linum usitatissimum* L.) proteome at the level of seed cake (SC), fine flour—sieved a fraction below 250 µm (FF)—and protein concentrate (PC). The evaluation was performed on three oilseed flax cultivars (Agriol, Raciol, and Libra) with different levels of α-linolenic acid content using LC-MS/MS (shotgun proteomics) analysis, which was finalized by database searching using the NCBI protein database for *Linum usitatissimum* and related species. A total of 2560 protein groups (PGs) were identified, and their relative abundance was calculated. A set of 33 quantitatively most significant PGs was selected for further characterization. The selected PGs were divided into four classes—seed storage proteins (11S globulins and conlinins), oleosins, defense- and stress-related proteins, and other major proteins (mainly including enzymes). Seed storage proteins were found to be the most abundant proteins. Specifically, 11S globulins accounted for 41–44% of SC proteins, 40–46% of FF proteins, and 72–84% of PC proteins, depending on the cultivar. Conlinins (2S albumins) were the most abundant in FF, ranging from 10 to 13% (depending on cultivar). The second most important class from the point of relative abundance was oleosins, which were represented in SC and FF in the range of 2.1–3.8%, but only 0.36–1.20% in PC. Surprisingly, a relatively high abundance of chitinase was found in flax products as a protein related to defence and stress reactions.

## 1. Introduction

Flaxseeds (*Linum usitatissimum* L.) have been used as human food or animal feed since historical times [1]. Flaxseed contains 30–46% oil, 18–30% protein, 20–35% fibre (about 10% is soluble fibre), 3–4% ash, and 4–8% moisture [1,2,3,4,5]. Minor constituents include cyanogenic glycosides, phytic acid, polyphenols, linatine, lignans (phytoestrogens), cyclolinopeptides, vitamins, selenium, and cadmium or trypsin inhibitors [2]. Due to their unique composition, flax seeds are classified as a functional food often referred to as a superfood [6,7]. Flaxseed oil is rich in essential polyunsaturated fatty acids, with n-3 alpha-linolenic acid (ALA; C18:3) 47–55% and n-6 linoleic acid (C18:2) represented by ≈15% in the oil of traditional cultivars [8]. Flaxseed oil with predominance of n-3 linolenic acid is not only used for human consumption but also, due to its drying properties, in industrial applications such as production of paints and varnishes, linoleum, oleo-chemicals, etc. [1,4,9,10]. The iodine value, which indicates the unsaturation level of the oils, is as high as 205 g I_2_/g oil [11]. There are few vegetable oil plants that provide oil with a similar level of iodine value, e.g., Dragon’s head oil (*Lallemantia iberica*) [12].

After mechanical extraction of the oil by pressing, flaxseed cake is a valuable material not only for farm animal feed, but also for the production of flaxseed flour, water-soluble fibre and protein concentrates [5,13,14,15,16], which, due to their functional properties, can be applied in a number of food applications in the production of quality foods that provide considerable health benefits to consumers [7,16]. High-protein flaxseed products can be used to enrich food products with nutritionally valuable components, improving their properties, including consistency, taste, and aroma. Flaxseed flours can be used in bakery products (bread, cookies, bars, biscuits, etc.), in pasta, but also, for example, in salad dressings or meat products [7]. Flaxseed protein flours have great potential in producing gluten-free products [17]. On the other hand, the food processing of flaxseeds and flaxseed products (e.g., roasting) can lead to several positive and negative changes, which also affect the protein and should be further monitored [18].

Flaxseed proteins in the form of concentrates or isolates are most commonly produced from defatted flaxseed meal stripped of water-soluble mucilage (which complicates protein concentration). First, alkaline solubilisation of the proteins is performed and then the proteins are precipitated by isoelectric precipitation [5,14,15]. The disadvantage of this most widely used method can be the varying extent of denaturation of flaxseed proteins (decreasing their solubility and functional properties), which can be caused by a high pH value (over 9) during the solubilisation step and may be improved by optimising the process conditions or partially hydrolysing the obtained proteins, which also yields valuable biologically active peptides [19,20,21], or by using an alternative method of protein concentration, e.g., protein extraction using ionic strength or membrane techniques [3,15].

Flaxseed protein is rich in glutamic and aspartic acid (including their amides) and the basic amino acid arginine. The total content of essential amino acids is 34%; leucine and lysine are the limiting amino acids in flaxseed protein, especially in children’s diets [4,22]. Flaxseed proteins are divided into globulins (linin), which account for up to 80% of flaxseed proteins, and albumins (conlinin), which represent a minor component compared to globulins. Diverse information is given on the structure of linin or 11–12S protein. While Madhusudhan and Singh [23] present flaxseed major protein as a 12S protein with 5 non-identical subunits with a total molecular weight (MW) of 294 kDa, later papers describe an acidic (MW = 30.0–35.2 kDa) and basic (MW = 24.6 kDa) subunit in 11S flaxseed globulin [24] or present it as a protein with a total MW of 365 kDa, which, after separation on SDS-PAGE under reducing conditions, separates into bands with MWs of 20, 23, and 31 kDa [25]. Barvkar et al. [1], in their unique study of the proteomic profile of flaxseed, report 11S globulin (with the names legumin, glutenin type A or cupin) as the major protein. The second major protein group is the 1.6–2S albumin referred to as conlinin, which is composed of a single polypeptide chain of MW 16–18 kDa. Other important flaxseed proteins are reported to be the 7S storage protein, oleosins, late embryogenesis abundant (LEA) proteins and a number of metabolically important enzymes [1,26]. The importance of flaxseed proteins, especially 11S globulins and 2S albumins, in relation to food applications, human nutrition and medical applications has been sufficiently discussed in recent works [6,15,26,27,28].

Information on the flaxseed proteome is scarce, e.g., the work of Barvkar et al. [1], which deals with the evaluation of changes in the flaxseed proteome during its development, or the study of Klubicová et al. [29], which deals with changes in the flaxseed proteome in response to cultivation in the Chernobyl region under phytoremediation.

The evaluation of protein profiles of flaxseed products with different protein contents (seed cake, flour, protein concentrate), using a proteomic approach has not yet been performed or is not available. Similarly, information on the cultivar differences of these flaxseed products is not available. For these reasons, the results presented in this article can be considered to be new. The knowledge of the protein profile is not only important from a nutritional and technological point of view, but it is also important to better understand the physiology of mature seed, the influence genotype and method of processing on proteome (protein composition), and it enables more efficient verification of the addition of flaxseed products in food products or for monitoring the presence of potentially risky proteins.

The objectives of the present work were (a) to describe the protein profiles of flaxseed products using shotgun proteomics at the level of quantitative estimation (determination of relative abundance) of identified protein groups, (b) to evaluate the effects of the protein concentration approach used (sieving versus alkaline solubilisation/isoelectric precipitation) and the cultivar effect on the changes in the relative abundance of food-relevant flaxseed proteins such as 11S globulins, 2S albumins (conlinin), and oleosins.

## 2. Materials and Methods

### 2.1. Flaxseeds

Flaxseeds of three cultivars of oilseed flax (*Linum usitatissimum* L.) were grown at the experimental station of the Faculty of Agriculture and Technology of the University of South Bohemia in České Budějovice (GPS 48.9742 N, 14.4477 E; 380 m above sea level). These cultivars were Agriol (cultivar number LNU25748), yellow seed colour, low ALA content (Agritec Plant Research, Ltd., Šumperk, Czech Republic); Raciol (cultivar number LNU11435), yellow seed colour, medium ALA content (Agritec Plant Research, Ltd., Šumperk, Czech Republic); and Libra (cultivar number LNU19160), brown seed colour, high ALA content (Limagrain Nederland B.V., Rilland, The Netherlands). Flaxseed cultivars used were grown in 2020. The sowing of the crop took place on 7 April 2020. The soil was satisfactorily supplied with the main nutrients, and no nitrogen in mineral fertilisers was applied. The crop was treated against weeds with the herbicide Glean 75 WG (chlorsulfuron) at a 15 g/ha dose. The seeds were harvested on 27 August 2020 using Wintersteiger Nursery Master Elite small plot research combine harvester (Wintersteiger Seedmech, GmbH, Ried im Innkreis, Austria). After harvesting, the flaxseeds were cleaned of impurities, including moisture-increasing contaminants. Until analyses were carried out, seeds were stored in the dark in paper bags inside polypropylene bags at 4 °C and 42% relative humidity.

### 2.2. Preparation of Flaxseed Cake, Flour, and Protein Concentrate

Preparation of analysed flaxseed products is shown schematically in Figure 1. The seeds were de-mucified by water soaking in two steps. In the first step, 200 g of seeds was mixed with 800 mL of water preheated to 50 °C. The mucilage was extracted for 1 h and removed from the mixture. In the second step, the seeds were soaked in 600 mL of water for 1 h at 50 °C, while the second step was performed twice in total. Subsequently, the wet seeds were dried on a sieve for 2 days at 50 °C. Afterwards, the seeds were subjected to oil pressing using a Yoda YD-ZY-02A domestic press (Yoda, Jinjiang, China) using the “flaxseed” program. The obtained cake was subjected to repeated pressing. Subsequently, the defatted cake was ground using a Grindomix GM 200 knife mill (Retsch, Haan, Germany) at 10,000 rpm for 1 min to obtain seed cake (SC). The flour was further sieved on a stainless-steel sieve (Preciselect, Tišnov-Malhostovice, Czech Republic) with a mesh size of 250 µm to produce fine flaxseed flour (FF). FF was used as a starting material for the preparation of protein concentrate (PC) through alkaline solubilisation followed by isoelectric precipitation (AE/IEP). Protein solubilization (extraction) was carried out as follows: deionized water was added to FF in a centrifuge plastic tube at a ratio of 1:10 (*w*/*v*), the mixture was mixed properly and then the reaction was adjusted to pH 8.5 using 1M NaOH, the total extraction was carried out at room temperature for 2 h with continuous control and pH adjustment. Subsequently, centrifugation was performed at RPM 4500 for 10 min and at RT on a Rotina 420 R centrifuge (Hettich, Tuttlingen, Germany). The proteins in the obtained supernatant were then subjected to precipitation by decreasing the pH to a value of 4.5 using 2M HCl. After centrifugation (under the same conditions as above), the obtained pellets were dissolved in deionized water, the pH was adjusted to 7.0 using 1M NaOH, the sample was frozen (−20 °C) and then freeze-dried using an Alpha 1–4 LSC lyophilizer (Martin Christ, Osterode am Harz, Germany) at a pressure of 0.520 mBar, and a temperature of −50 °C for 72 h. The obtained lyophilizate was homogenized to a powder representing the protein concentrate (PC).

### 2.3. Proximate Composition Analysis

All the proximate analyses were determined according to Bárta et al. [30]. Moisture was determined gravimetrically by drying the samples in an oven to constant weight at 105 °C for 3 h. The protein content was determined by the modified Dumas combustion method using a rapid N cube (N/Protein analysis) instrument (Elementar Analysen System, Langenselbold, Germany). The protein content was calculated as the nitrogen content multiplied by a factor of 6.25. The fat content was measured using the Soxhlet extraction method using an ANKOM XT 10 Extractor (ANKOM Technology, Macedon, NY, USA) according to the manufacturer’s manual. Petroleum ether was used as an extraction reagent. The fat content was calculated from the weight differences of the sample before and after extraction. The ash content was measured as the total amount of inorganic residues remaining after the high heat incineration of analysed samples at 550 °C for 5 h in a muffle furnace (muffle furnace LE 09/11, LAC s.r.o., Židlochovice, Czech Republic). The content of carbohydrates was determined as the remaining residue of the total fresh matter (FM) after subtracting the sum of moisture, protein, fat, and ash. Each sample was analysed in triplicate for all analyses.

### 2.4. One-Dimensional Electrophoresis (SDS-PAGE)

The flaxseed product samples were extracted with SDS extraction buffer [0.065 M Tris-HCl, pH 6.8; 2% (*w*/*v*) SDS; 5% (*v*/*v*) 2-sulphanylethanol] in a ratio of 1:10 (*w*/*v*) at 4 °C for 4 h. Cooled dual vertical slab units SE 600 (Hoefer Scientific Instruments, Holliston, MA, USA) were used for protein separation. Protein samples were separated with a discontinuous gel system (4% stacking and 11% resolving gel) in reduction conditions (modified according to Laemmli [31] and Bárta et al. [32]. Each research variant was analysed in triplicate. Protein detection was performed using Coomassie Brilliant Blue R-250.

### 2.5. LC-MS/MS Analysis and Proteomic Data Processing

Flaxseed product samples were processed in triplicate in a similar way as hemp samples in Bárta et al. [32]. Proteins from flaxseed product samples were isolated using SDT buffer (4% SDS, 0.1M DTT, 0.1M Tris/HCl, pH 7.6) in an Eppendorf ThermoMixer C (120 min, 95 °C, 750 rpm). Subsequently, samples were centrifuged (15 min, 20,000× *g*). Then, the supernatants (containing ca 100 μg of total protein) were processed using filter-aided sample preparation (FASP) as described elsewhere [33] using 0.5 μg of trypsin (sequencing grade; Promega). Resulting peptides were analysed using LC-MS/MS.

Obtained peptide mixtures were analysed using LC-MS/MS using the UltiMate 3000 RSLCnano system coupled to Orbitrap Exploris 480 mass spectrometer (Thermo Fisher Scientific, Waltham, MA, USA). Tryptic digests were online concentrated and desalted on trapping column (Acclaim PepMap 100 C18, dimensions 300 μm ID, 5 mm long, 5 μm particles, Thermo Fisher Scientific, heated to 40 °C). Trapping column with peptides was washed with 0.1% FA; the peptides were eluted in backflush mode (350 nL·min^−1^) from the trapping column. Gradient elution (90 min, flow rate 250 nL·min^−1^, 3–37% of mobile phase B; mobile phase A: 0.1% FA in water; mobile phase B: 0.1% FA in 80% ACN) was applied for separation on an analytical column (EASY-Spray column, 75 μm ID, 500 mm long, 2 μm particles, Thermo Fisher Scientific, heated to 50 °C).

A data-independent acquisition mode (DIA) was selected for data acquisition. The *m*/*z* range of 350–1400 at a resolution of 60,000 (at *m*/*z* 200) and maximum injection time of 55 ms was used for the survey scan. The *m*/*z* range of 200–2000 at 30,000 resolution (maximum injection time 55 ms, 27% relative fragmentation energy) was selected for HCD MS/MS spectra collection. Overlapping windows scheme in *m*/z range from 400 to 1000 were used as isolation window placements.

DIA-NN software (version 1.8.1) [34] was employed for the processing of acquired DIA data. The library free mode was used for searches against modified cRAP database (based on http://www.thegpm.org/crap/ (accessed on 22 November 2018); 111 sequences in total) and whole *Linum* genus database (https://www.ncbi.nlm.nih.gov/ipg/?term=txid4005[Organism:exp] (accessed on 30 September 2023, number of protein sequences: 125,854). Carbamidomethylation as fixed modification, with no optional modification and trypsin/P enzyme with 1 allowed missed cleavage and peptide length 7–30, were set for preparation of the library. The false discovery rate (FDR) control was set to 1% FDR. The initial test searches were performed to set MS1 and MS2 accuracies and scan window parameters (median value from all samples ascertained parameter values). MBR was switched on.

The main report file generated using the DIA-NN was further processed using the software container environment (https://github.com/OmicsWorkflows/KNIME_docker_vnc (accessed on 4 October 2023)), version 4.6.3a. The processing workflow is available upon request. Briefly, it covered (a) removal of low-quality precursors and contaminant protein groups, (b) estimation of intensity-based absolute quantities (iBAQ values) using DIA-TPA [35] algorithm using normalized precursor intensities, (c) relative iBAQ (riBAQ) values calculation for individual protein groups within individual sample replicates. DIA-TPA algorithm implementation calculated iBAQ (and following riBAQ) values for all protein groups having at least one proteotypic peptide and or protein groups without any proteotypic peptide, but where all proteins of the given protein group were reported only together.

The description of the identified protein groups presented in the results section of the article (Section 3.2) was based on information available in the NCBI or UniProt KB databases.

### 2.6. Statistical Analysis

Obtained data were analysed using software Statistica 12 (StatSoft Power Solutions Inc., Palo Alto, CA, USA). Two-way ANOVA (analyses of variance) and the variance components method was used for data evaluation. The means were compared using the Tukey HSD test. Differences between the variants were considered significant at *p* < 0.05 unless stated otherwise.

## 3. Results and Discussion

### 3.1. Proximate Composition of Flaxseed Products

The proximate chemical composition of the analysed flaxseeds (WS) and the flax products derived from them is given in Table 1. The individual processing steps from seed to flaxseed protein concentrate (WS, SC, FF, PC), as shown in Figure 1, lead to a gradual relative increase in the average protein content, from 18% (WS), over 30% (SC), and 36% (FF) to 70% or more (PC) of the three oilseed flax cultivars evaluated. The differences between the cultivars are noticeable, but the trend and the ratio of the gradual increase are similar for all the cultivars evaluated. The WS fat content of all three cultivars has decreased from the original 38–40% to below 10% in SC due to cold pressing (up to 50 °C). During sieving SC using a sieve with a mesh size of 250 µm, a fractionation effect occurred resulting in increased protein and fat content but less carbohydrates in flaxseed flour (FF), compared to the original SC.

The production of oil, flour, or protein from flaxseed is more difficult than for other oilseeds because of water-soluble mucilages (gums), which complicate processing, especially ‘wet’ processing. The presence of these mucilages and unsophisticated isolation methods have been the main reasons that have hindered the wider dissemination of flax protein [25,36]. Various methods have been tried to remove (or reduce) the mucilage content in flaxseeds, including the use of degradative enzymes and extraction using water at different temperatures [14,37]. Kaushik et al. [14] tested the effect of temperature during water extraction of the mucilage, prior to the actual processing of the seed. The authors concluded that water extraction at 60 °C can yield flaxseed protein isolate with purity of over 90%. Sieving is also important in changing the proportions of the content of the moulded fibre. The finer fractions of the flour (under a sieve with a size of about 250 µm) have a higher protein (and possibly fat) content and a lower carbohydrate (especially fibre) content, as sieving removes a greater proportion of the coarse fragments that represent the surface layers of the seed or hull [30].

### 3.2. Description of Main Identified Protein Groups in Flaxseed Products

The protein profiles of flaxseed products derived from seeds after oil pressing of the three cultivars separated using SDS-PAGE are shown in Figure 2.

A typical electrophoretic profile of oilseed proteins separated under denaturing and reducing conditions [27,30] is evident in all samples and it agrees with reported studies dealing with flaxseed proteins [1,5,25,38,39]. The proteins can be divided into four MW regions: the protein with an MW of 48 kD (region A), the acidic subunits of the 11S globulin monomer (region B), the basic subunits of the 11S globulin monomer (region C), and the region of 2S albumin conlinin and oleosins (region D). There are minor differences between the seed protein profiles of the evaluated cultivars. The PC profiles of all three cultivars are slightly simpler with a predominance of protein bands in the regions of the acidic and basic subunits of the 11S protein monomer (globulin). This suggests that the chosen alkaline solubilisation (extraction) procedure at pH 8.5 and the subsequent precipitation of the extracted proteins caused by decreasing the pH to 4.5 leads to a higher relative abundance of 11S globulins (discussed more in Section 3.3). However, this suggestion can be taken with some reserve because the absolute amount of protein in the samples loaded on the electrophoretic gel differed.

After LC-MS/MS analysis of the peptide mixtures of the samples, the obtained data were subjected to qualitative and quantitative evaluation using available databases of primary protein sequences. Considering the unavailability of reference proteome for flax, the NCBI database (LINUM taxonomy) was primarily used, with the help of which 2560 protein groups (PGs) were identified (see Appendix A). For the assessment of relative abundance (determined from measured intensities, riBAQ values), the set of 2560 identified PGs was assumed to be 100%. For detailed evaluation, a set of 33 PGs with high relative abundance in the total protein profile (PGs above 0.5%) or belonging to important flaxseed proteins was selected. The PGs items in this set were divided into four classes (Table 2) into seed storage proteins, oleosins, proteins associated with defence and stress, and other selected important proteins.

#### 3.2.1. Seed Storage Proteins

The seed storage proteins (SSPs) in flaxseed are mainly classified as 11S (12S) globulins and 2S albumins [3,16,25,27]. The globulin fraction is often referred as linin [3]; however, PGs are not listed under this name in the NCBI and UniProt KB databases. In addition to 11S globulins, we were able to find five items in the set of identified PGs (Table 2). In terms of relative abundance, the most significant items are CAI0558954.1, CAI0432378.1, and CAI0558846.1, which are designated as “cupin 11S legumins” or “11S globulin/legumin seed storage proteins” or “11S globulin-like proteins”.

According to other authors [3,25,28], these are polymeric proteins with MW ≈ 300 kDa. The PGs we have found have primary polypeptide sequences of 367–536 amino acids in length, from which two sub-polypeptide chains, an acidic and a basic one, are cut during post-translational processing, similarly to proteins of other oilseeds [40], which then give rise to the final monomer by their association via a disulfide bond. Due to their high relative abundance, 11S globulins have a major impact on the amino acid composition of flaxseed proteins. The nutritional quality of flaxseed proteins is close to that of soya and canola proteins. Their essential amino acid index is reported at 69, whereas values of 79 and 75 are reported for soya and canola proteins, respectively. The significantly represented amino acids of flaxseed proteins include arginine, aspartic, and glutamic acids (including their amides) [3]. Flaxseed protein has a lysine to arginine ratio of about 0.25 and, therefore, has less lipidemic and atherogenic effects than, for example, soya protein (0.71), milk caseinate (2.15), or whey protein (5.38), so it is referred to as a “better” protein for the health of cardiac patients [14].

A second important part of flax SSPs are the 2S albumins, commonly referred to as conlinins [1,26,41]. In the set of identified proteins, two PGs (CAC94011.1, CAC94010.1) were found with the name conlinin and the described activity of the storage protein (Table 2). According to the accession number and the assigned GI number, these items are identical to the two proteins found by Liu et al. [41] in their work. These authors studied conlinin for its involvement in the emulsification activity of the flaxseed soluble gum (FSG) fraction, to which conlinins were co-transferred during the hot water isolation of FSG. The conlinins were found to be significantly involved in increasing the emulsification activity and stability of the emulsions formed. These are proteins that have been described in databases based on the cDNA sequence of two conlinin genes, cnl1 and cnl2 [42]. Earlier work [23] describes conlinins as proteins with a sedimentation coefficient of 1.6S and a MW of 15–18 kDa, while Liu et al. [41] found a MW by 2D-PAGE of around 11 kDa; however, the UniPro KB database lists a predicted MW of 19.063 kDa (169 amino-acids) for Q8LPD4 (CAC94011.1) and a predicted MW of 19.012 kDa (168 amino-acids) for Q8LPD3 (CAC94011.1).

Diverse physiological functions are generally reported for seminal 2S albumins. Souza [43] in his work states that they are not only proteins with a storage function as a source of indispensable amino acids (especially cysteine), but due to their alpha–helical structure and positively charged amino acid residues, they are often characterized by antimicrobial and antifungal activity. The more significant use of 2S albumins in food and biotechnological applications is hampered by their relatively high allergenicity [44]. Additionally, flaxseed polypeptides with MW from 20 to 38 kDa (or more), e.g., malate dehydrogenase, are suspected of allergenicity, too [3]. Allergies to flaxseed are rarely reported. However, Bueno-Díaz et al. [6] described recently the allergy of five patients to flaxseed and 2S albumin was the main allergen, and 11S globulin was a potential allergen.

In addition to 11S globulins and 2S albumins, other proteins of similar function have been reported in flaxseed. For example, Barvkar et al. [1] described glutelins, a 48 kDa proteins, and 7S globulins in flaxseeds. Globulins 7S were detected in seeds of other oilseed crops including hemp [32]. No proteins of this type were found in our study.

#### 3.2.2. Oleosins

Oleosins (OLs) are proteins with MW ranging from 15 to 26 kDa [45,46] that participate in the structure of oil bodies (oleosomes). In oilseeds, oleosomes are mainly localized in the cotyledon’s cells, where they form structures with diameters between 0.2 and 2 µm [47]. Oleosomes are composed of a monolayer membrane of phospholipids reinforced not only by OLs but also by other types of proteins such as caleosins or steroleocins [47,48,49]. Typically, the oleosome isolated from oil bodies contains about 94–98% neutral lipids, 0.6–2% phospholipids and 0.6% protein [49].

The OLs are among the most quantitatively important proteins in oilseeds. We found a total of six oleosin PGs, whose relative abundance in the seed extracts of the three cultivars evaluated ranged from 2.12 to 2.86% in analysed flaxseed products. Four PGs were the “oleosin low molecular weight isoforms” (MW in the range 14.991–16.039 kDa) and two the “oleosin high molecular weight isoforms” (MW 18.643 kDa or 18.712 kDa). OLs relative abundance in flaxseeds is lower compared to hempseeds (8%) based on the results of the recent study by Bárta et al. [32].

#### 3.2.3. Defence- and Stress-Related Proteins

Within the PGs significantly related to flaxseed defence and its response to stress, we detected the enzyme chitinase, late embryogenesis abundant protein, heat shock proteins, and trypsin inhibitor. We observed a surprisingly high relative abundance of chitinase (represented by two PGs WMZ41542.1 and AAW31878.1) ranging from 6 to 10% in the flaxseed protein pool. Klubicová et al. [29] also found chitinase in mature flaxseeds harvested from remediated plots in the Chernobyl region, but at a significantly lower abundance of 0.3% (calculated as a percentage of spot volume in the evaluation of the information obtained on the gel after two-dimensional electrophoresis).

Among the proteins associated with the stress responses of the maturing seed, a PG with the designation “Late embryogenesis abundant (LEA) group 1—embryo development” (AMY26620.1) can be included. LEA proteins have been found in various organisms, including plants. In seed tissues, they act as protein and membrane protectors against drought stress and water deprivation [50].

The PG with the designation “Sequence 3353 from patent US 9878004” (accession number AVY09180.1) was found to be a quantitatively significant item (relative abundance of about 0.9%). Polypeptide of this PG consists of 158 amino acids, and the predicted MW is 17.99 kDa. The function of this PG in flaxseeds is unclear; however, a detailed search of the NCBI and UniProt KB databases revealed a high degree of sequence similarity (around 80%) to small heat shock protein (SHSP) type proteins. SHSPs are found in both prokaryotic and eukaryotic cells and are among the chaperones, and are generally part of the quality control mechanism of cellular proteins with a line of defence function against conditions that threaten the cellular proteome [51]. In plant seeds, the action of SHSPs is not only related to protection against heat stress. The expression of SHSPs in seeds occurs during the ripening process, leading to their accumulation in dry seeds, but their presence disappears after seed germination [52].

A significant finding is the presence of an item (1DWM_A) that forms part of a trypsin inhibitor. On the one hand, protease inhibitors are part of the natural defence of plants against harmful agents, which can be used biotechnologically or in plant protection [2,53], on the other hand, they can have adverse effects on human digestion and utilization of consumed proteins [54].

#### 3.2.4. Other Selected Important Proteins

Among other proteins significant in their relative abundance are enzymes metabolically relevant to mature seed physiology, e.g., glyceraldehyde-3-phosphate dehydrogenase, fructose-bisphosphate aldolase and malate dehydrogenase, whose presence in flaxseeds is also confirmed in the work of Barvkar et al. [1]. Among the significant PGs found in the flax products we analysed is AVY09175.1, which is probably related to hydroxysteroid dehydrogenase activity and oleosome (oil body) membrane proteins [55]. PGs with significant relative abundance in flax seeds also include allene oxide synthase (P48417.1) with a relative abundance of about 0.6%, which is likely involved in fatty acid biosynthesis, and translational elongation factor (CAI0411675.1) with a relative abundance of about 1%.

### 3.3. Influence of Cultivar and Concentration Method on Relative Abundance of Main Protein Groups

The relative abundance of individual proteins contained in flaxseed products can be influenced by several factors. Our study evaluated the influence of protein concentration methods producing different flaxseed products (FF and PC) in combination with the influence of cultivars. In terms of evaluating the influence of the concentration method factor, two methods used were chosen as examples of dry and wet processes to increase the protein concentration in products [16,30,56]. The dry process is represented by sieving milled SC through a sieve with a mesh size of 250 µm, while the wet process is represented by the AE/IEP combination. The 250 μm mesh size sieve was chosen, as in our work of Bárta et al. [30], based on our previous experience when we tested sieves with mesh sizes ranging from 180 to 710 μm. The 250 μm mesh sieve provided the best result regarding the protein concentration effect and practical advantages (the sieve did not clog). Analysis of the data set of selected PGs revealed that for 30 PGs out of 33 in total, the concentration method factor had a significantly greater influence than the cultivar factor.

The estimation of the relative abundance of the PGs for the flaxseed products derived from the three cultivars is given in Table 3. From the results, it is evident that SSPs have the highest relative abundance in the flaxseed protein pool. The PGs (especially items CAI0558954.1, CAI0432378.1, CAI0558846.1; CAI0434421.1), which represent 11S globulin or linin, respectively, occupy a relative representation in SC for all three cultivars in the range of 41–44%. Most authors report higher values, e.g., Bekhit et al. [4] report a relative representation of 64–66% for 11S globulin, while another study [7] reports a relative abundance of 70–85% for globulins. The difference may be explained by the fact that in most papers the abundance of globulins is reported as the abundance of the protein fraction soluble in salt solution, which may also contain other proteins. On the other hand, the 7S protein fraction, which is assigned to globulins [1], was not found in our samples.

The wet process of protein concentration using AE/IEP led to a significant increase in the relative abundance of 11S proteins in PC samples (Table 3, Figure 3). This method increased the representation of 11S globulins from a level of 40–46% to a level of 72–84% for the materials of all three cultivars. The effect of cultivar on the concentration of 11S globulins was less significant for the most represented items and was estimated by statistical analysis (variance component) to be 3–11%, while the effect of the concentration method was in the range of 79–95%. The effect of sieving the SC on the concentration of 11S globulins was ambiguous depending on the cultivar (see Figure 3). Alkaline solubilisation (extraction) allows the transition of water-insoluble globulins to the soluble state due to the prevailing isoelectric point (pI) values of globulin proteins. The pI values for 11S flaxseed globulins are reported to be around pH 4.5 [14]. Considering the known fact that proteins are insoluble in environments equal to or close to their pI, alkaline extraction in the pH range of 7.5–9.0 (12.0) allows sufficient “distance” of proteins from their pI value and conversion to a soluble state [56,57]. Increasing the concentration of the alkaline medium, and thus higher pH during extraction, generally increases the yield of proteins [57,58], but also increases the risk of protein damage—denaturation, protein hydrolysis, amino acid racemization, and loss of essential amino acids, including the formation of lysinoalanine with nephrotoxic activity [58,59]. The low residual fat content of the original flour or its defatting with organic solvent is also important, as pH adjustment through hydroxides leads to soap formation when reacting with fat.

The fraction of conlinins (2S albumin) was found in the SC of the evaluated cultivars at the level of 7–9% (Table 3), which is lower than have been reported by other authors. Often, the albumin fraction is reported at 15–30% [5,15]. As can be seen from Figure 3, for conlinins, mechanical sieving (increase from the original 7–9% to 10–13%) has a higher concentration effect than AE/IEP (decrease to 7.5–11%). Sieving resulted in an increase in relative abundance for two cultivars (Agriol, Libra), but AE/IEP resulted in a very low abundance of OLs in PC for the class OLs.

In the case of chitinase (item WMZ41542.1) and most other PGs of the defence- and stress-related proteins class (items AMY26620.1, CAI0408810.1, 1DWM_A), sieving and AE/IEP leads to a decrease in their abundance in FF and PC. This may be explained by the fact that proteins related to defence and stress reactions are present in higher concentrations in the seed coat and other surface layers of the seed tissues, where they are “closer” to factors attacking or affecting the seed. Sieving with a 250 µm mesh size sieve leads to the separation of coarser particles from the FF. These particles are generated during milling from the surface layers of the seed coat.

For other selected important proteins (mainly quantitatively important enzymes), no significant difference in their relative abundance in SC and FF was found, but a significant decrease in their abundance in PC was observed. Thus, these types of proteins may be found in higher concentrations in the residual flour after the solubilisation phase of the AE/IEP procedure (in the pellet after centrifugation of the extraction mixture) and this material may be an interesting source for their recovery and possible use in biotechnological processes.

From the point of view of food, nutritional and medical applications of flaxseed proteins, the major fractions of proteins such as 11S globulins, 2S albumins and oleosins, are particularly important. From the results presented above, it is evident that mechanical sieving leads to an increase in the relative abundance of 2S albumins in the product. On the contrary, the AE/IEP method significantly increases the relative abundance of 11S globulins, so that the obtained PC can have up to twice higher abundance than SC or FF. This is important information when practical use of their functional properties is required. Kaushik et al. [14] reported that flaxseed protein isolate (also obtained using the AE/IEP method) excels in water binding, has a stabilizing effect on emulsions formed with sunflower oil and, as discussed in Section 3.2, it may be a “better” protein for cardiac patients due to its low lysine to arginine ratio (only 0.25). Flaxseed protein concentrate may be a good source of bioactive peptides having antibacterial, antioxidant, and antidiabetic effects as well as an angiotensin-converting enzyme inhibition effect due to the amino acid sequence of 11S globulin [15]. Merker et al. [28] highlighted the potential of flaxseed proteins including 11S globulins to produce peptides with an anticarcinogenic effect.

As already mentioned, the effect of cultivar on the different abundances of the PGs evaluated in flaxseed products was lower than the effect of the concentration method. However, for PCs, significant differences were found between cultivars in the relative abundance of 11S globulins. While the high ALA cultivar Libra had a relative abundance of 11S globulins (in total) of 71.73%, the low ALA cultivar Agriol had a relative abundance of 84.02%. These findings suggest that cultivar selection may significantly influence the representation of these proteins in PC, but this idea needs to be further developed in follow-up research by obtaining data from a larger set of cultivars.

## 4. Conclusions

A total of 2560 protein groups were found within the evaluated flaxseed products (seed cake, fine flour, protein concentrate) prepared from the three oilseed flax cultivars despite the incomplete sequence information for *Linum usitatissimum*. Subsequently, incomplete annotations made creating a functional description rather difficult for most of the identified protein groups.

The only 33 protein groups that were selected for detailed description were identified to have a relative abundance within the protein pool of 69–95%. The conditions combining alkaline solubilisation (at pH 8.5) and subsequent isoelectric precipitation of the proteins (at pH 4.5) significantly increased the relative abundance of 11S globulin-type proteins (from 41–44% to 72–84%) in the resulting protein concentrate, which have high potential for food applications due to their properties and amino acid composition.

Mechanical sieving of milled cake increased the relative amount of 2S albumin in the fine flour (up to the value of 9–13%, depending on the cultivar). Another important protein class, oleosins, was more present in seed cake or fine flour, depending on the cultivar, ranging from 2–4%. Both protein classes (2S albumin and oleosins) have been shown to decrease their abundance in protein concentrate compared to 11S globulins. A surprising and interesting finding is the high relative abundance of chitinase (up to 10% in seed cake and fine flour, respectively), which is part of the plant defence system against microorganisms. The majority of the other protein groups evaluated were enzymes (e.g., glyceraldehyde 3-phosphate dehydrogenase, fructose-bisphosphate aldolase, or allene oxide synthase), which crossed over to protein concentrate in only a small proportion.

The effect of cultivar on the relative abundance of the proteins evaluated is not negligible, but compared to the method of the protein concentration is significantly lower for the vast majority of identified protein groups. The low ALA cultivar Agriol had the highest relative abundance of 11S globulins (84%), whereas the high ALA cultivar Libra had the lowest relative abundance of 11S globulins (almost 72%). The selection of a suitable cultivar could be an important factor to obtain a protein concentrate with the highest abundance of 11S globulins.

## Figures and Tables

**Figure 1 foods-13-01288-f001:**
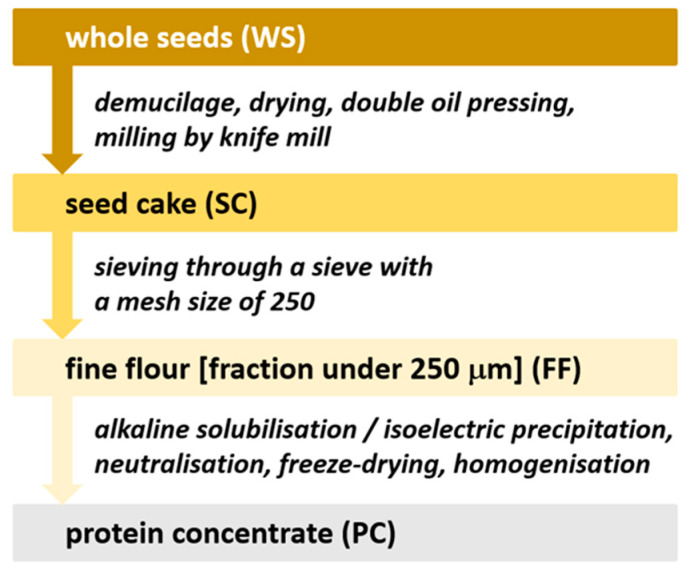
Preparation of analysed flaxseed products from whole seeds.

**Figure 2 foods-13-01288-f002:**
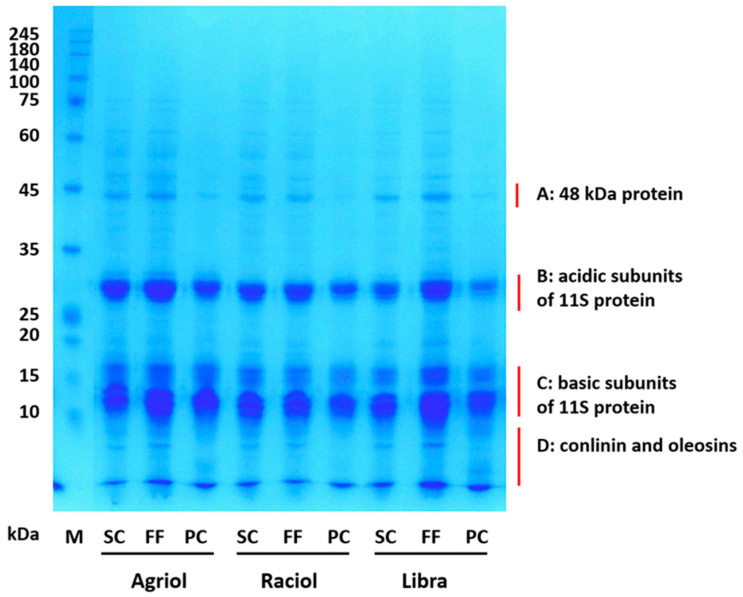
1D SDS-PAGE profiles of flaxseed products (seed cake—SC, fine flour—FF, protein concentrate—PC) from three oilseed flax cultivars (Agriol, Raciol, Libra). Note: M—molecular mass standard (RotiMark Tricolor [8271.1], Carl Roth GmbH + Co. KG, Karlsruhe, Germany).

**Figure 3 foods-13-01288-f003:**
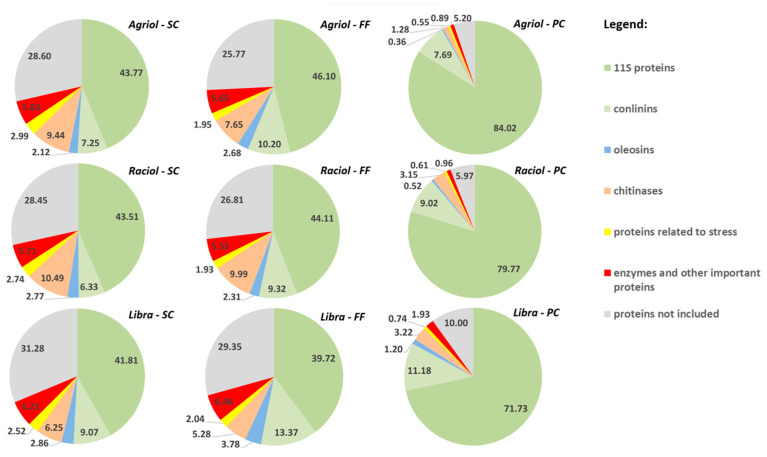
Relative proportions of protein classes depending on the flaxseed product (seed cake—SC, fine flour—FF, protein concentrate—PC) and cultivar.

**Table 1 foods-13-01288-t001:** Proximate composition of whole flaxseeds (WS) and derived flaxseed products—seed cake (SC), fine flour (FF), and protein concentrate (PC).

Component	Cultivar	WS	SC	FF	PC
Protein (% FM)	Agriol	18.02 ± 0.58 Ad	30.19 ± 1.24 Ac	35.81 ± 0.50 Ab	78.34 ± 0.52 Aa
	Raciol	17.06 ± 0.66 Ad	29.27 ± 0.59 Ac	33.87 ± 0.89 Ab	70.09 ± 1.63 Ba
	Libra	17.84 ± 1.04 Ad	30.42 ± 1.04 Ac	36.02 ± 0.22 Ab	70.87 ± 1.28 Ba
Fat (% FM)	Agriol	37.76 ± 0.61 Ba	8.53 ± 0.47 Ac	10.94 ± 0.04 Ab	2.41 ± 0.77 Bd
	Raciol	38.24 ± 0.08 Ba	9.46 ± 0.16 Ac	12.04 ± 1.09 Ab	2.42 ± 0.36 Bd
	Libra	40.76 ± 0.54 Aa	9.22 ±1.20 Ac	12.21 ± 1.22 Ab	5.04 ± 1.27 Ad
Carbohydrates (% FM)	Agriol	34.27 ± 0.47 Ac	47.28 ± 1.23 Aa	38.75 ± 0.66 Ab	9.78 ± 0.54 Cd
	Raciol	34.86 ± 0.67 Ac	48.01 ± 0.31 Aa	39.68 ± 0.70 Ab	19.41 ± 2.15 Ad
	Libra	31.90 ± 1.69 Ac	46.29 ± 0.91 Aa	36.87 ± 0.82 Ab	15.99 ± 0.32 Bd
Ash (% FM)	Agriol	3.43 ± 0.12 Ac	5.36 ± 0.12 Bb	6.28 ± 0.03 Ba	2.71 ± 0.12 Ad
	Raciol	3.52 ± 0.09 Ac	5.27 ± 0.08 Bb	6.28 ± 0.02 Ba	2.72 ± 0.08 Ad
	Libra	3.52 ± 0.09 Ac	5.78 ± 0.13 Ab	7.01 ± 0.04 Aa	2.82 ± 0.15 Ad
Moisture (% FM)	Agriol	6.52 ± 0.04 Ab	8.63 ± 0.24 Aa	8.22 ± 0.21 Aa	6.76 ± 0.86 Ab
	Raciol	6.32 ± 0.05 Ab	7.99 ± 0.32 Aa	8.12 ± 0.13 Aa	5.36 ± 1.03 Bb
	Libra	5.98 ± 0.08 Ab	8.29 ± 0.14 Aa	7.89 ± 0.24 Aa	5.27 ± 0.15 Bb

Note: FM—fresh matter; different capital letters in the columns for individual component values indicate a statistically significant cultivar difference at *p* < 0.05 (Tukey HSD test); different lower-case letters for the component values in the rows indicate a statistically significant difference between the flaxseed products of the respective cultivar at *p* < 0.05 (Tukey HSD test).

**Table 2 foods-13-01288-t002:** Description of selected protein groups found by proteomic processing of flaxseed products.

Accession	Name of Protein Groups	Description from UniProt KB/(Available Information from NCBI)
Seed storage proteins		
CAI0558954.1	unnamed protein product; LINTE	(cupin 11S legumin N/11S legumin seed storage globulin/N-terminal cupin)
CAI0432378.1	unnamed protein product; LINTE	(11S globulin-like/11S globulin seed storage protein)
CAI0558846.1	unnamed protein product; LINTE	(cupin_RmlC-like protein)
CAI0434421.1	unnamed protein product; LINTE	(cupin_RmlC-like protein) 11S globulin
CAI0418748.1	unnamed protein product; LINTE	(cupin_RmlC-like protein)
CAC94011.1	conlinin; LINUS	2S seed storage albumins family; nutrient reservoir activity
CAC94010.1	conlinin; LINUS	2S seed storage albumins family; nutrient reservoir activity
Oleosins		
ABB01617.1	oleosin low molecular weight isoform; LINUS	oleosin family; membrane; monolayer-surrounded lipid storage body
ABB01620.1	oleosin low molecular weight isoform, partial; LINUS	oleosin family; membrane; monolayer-surrounded lipid storage body
ABB01616.1	oleosin high molecular weight isoform; LINUS	oleosin family; membrane; monolayer-surrounded lipid storage body
ABB01624.1	oleosin high molecular weight isoform; LINUS	oleosin family; membrane; monolayer-surrounded lipid storage body
ABB01618.1	oleosin low molecular weight isoform; LINUS	oleosin family; membrane; monolayer-surrounded lipid storage body
ABB01619.1	oleosin low molecular weight isoform, partial; LINUS	oleosin family; membrane; monolayer-surrounded lipid storage body
Defence and stress-related proteins		
WMZ41542.1	chitinase; LINUS	(chitinase; involved in somatic embryogenesis in flax)
AAW31878.1	chitinase, partial; LINUS	chitinase activity; chitin binding; cell wall macromolecule catabolic process; defense response
AMY26620.1	late embryogenesis abundant (lea) group 1-embryo development; LINUS	LEA type 1 family; embryo development ending in seed dormancy
AVY09180.1	Sequence 3353 from patent US 9878004; UNKNOWN	(17.3 kDa class I heat shock protein)
CAI0408810.1	unnamed protein product; LINTE	(luminal-binding protein 5; heat shock 70 kDa protein)
1DWM_A	Chain A, Linum usitatissimum trypsin inhibitor; LINUS	(Serine proteinase inhibitor class)
Other selected important proteins		
AVY09175.1	Sequence 3348 from patent US 9878004; UNKNOWN	(11-beta-hydroxysteroid dehydrogenase)
CAI0407981.1	unnamed protein product; LINTE	(glucose and ribitol dehydrogenase/NADPH-dependent aldehyde reductase 1)
CAF22093.1	glyceraldehyde 3-phosphate dehydrogenase, partial; LINUS	glyceraldehyde-3-phosphate dehydrogenase family; oxidoreductase
BAL41455.1	fructose-bisphosphate aldolase 1, partial; LINGR	class I fructose-bisphosphate aldolase family; fructose-bisphosphate aldolase activity; glycolytic process
P48417.1	Allene oxide synthase, chloroplastic; LINUS	allene oxide synthase; activity; fatty acid biosynthetic process
ABM64783.1	cell wall glycosidase, partial; LINUS	hydrolase activity, hydrolyzing O-glycosyl compounds; carbohydrate metabolic process
CAI0384355.1	unnamed protein product; LINTE	(copper/zinc superoxide dismutase)
CAI0431450.1	unnamed protein product; LINTE	(manganese superoxide dismutase)
CAI0412769.1	unnamed protein product; LINTE	(linoleate 13S-lipoxygenase 2-1, chloroplastic; Lipoxygenase)
AVY09204.1	Sequence 3377 from patent US 9878004; UNKNOWN	(hypothetical protein DKX38_015183; malate dehydrogenase)
CAI0440995.1	unnamed protein product; LINTE	(putative mitochondrial malate dehydrogenase; malate dehydrogenase, mitochondrial)
CAI0411675.1	unnamed protein product; LINTE	(elongation factor 1-alpha; translation elongation factor)
AGN56421.1	non-specific lipid transfer protein 1; LINUS	lipid binding; lipid transport; plant LTP family
CAI0414382.1	unnamed protein product; LINTE	(thaumatin like protein)

Note: LINUS—*Linum usitatissimum*; LINTE—*Linus tenue*; LINGR—*Linus grandiflorum*; UNKNOWN.

**Table 3 foods-13-01288-t003:** Relative abundance of selected protein groups in flaxseed products (seed cake—SC, fine flour—FF, protein concentrate—PC) of three oilseed flax cultivars expressed in relative % (mean ± std. deviation).

Accession	Protein Name		Agriol			Raciol			Libra	
		SC	FF	PC	SC	FF	PC	SC	FF	PC
Seed storage proteins										
CAI0558954.1	unnamed protein product; LINTE	29.13 c ± 1.30	30.44 c ± 0.32	53.37 a ± 0.45	31.09 bc ± 6.62	29.37 c ± 0.43	51.71 a ± 1.63	26.34 c ± 1.10	23.34 c ± 0.28	38.83 b ± 4.24
CAI0432378.1	unnamed protein product; LINTE	7.12 cd ± 0.14	8.06 cd ± 0.70	16.08 ab ± 0.83	6.39 d ± 0.77	7.94 cd ± 0.44	14.57 b ± 0.30	8.19 cd ± 0.93	8.89 c ± 0.19	16.88 a ± 0.95
CAI0558846.1	unnamed protein product; LINTE	6.69 b ± 0.15	6.42 b ± 0.29	11.86 a ± 0.55	5.21 b ± 0.43	5.69 b ± 0.26	10.91 a ± 0.28	6.37 b ± 0.68	6.30 b ± 0.10	13.23 a ± 0.20
CAI0434421.1	unnamed protein product; LINTE	0.82 bc ± 0.02	0.97 bc ± 0.05	2.44 a ± 0.10	0.65 c ± 0.15	0.92 bc ± 0.02	2.41 a ± 0.09	0.68 c ± 0.03	1.06 b ± 0.07	2.59 a ± 0.29
CAI0418748.1	unnamed protein product; LINTE	0.01 a ± 0.01	0.21 a ± 0.02	0.27 a ± 0.01	0.17 a ± 0.02	0.19 a ± 0.03	0.17 a ± 0.15	0.23 a ± 0.01	0.13 a ± 0.11	0.20 a ± 0.21
CAC94011.1	conlinin; LINUS	7.01 ab ± 1.09	9.62 ab ± 1.13	7.43 ab ± 1.41	5.90 b ± 1.29	9.01 ab ± 1.55	8.56 ab ± 0.93	7.72 ab ± 2.31	11.32 a ± 1.01	9.72 ab ± 2.51
CAC94010.1	conlinin; LINUS	0.24 c ± 0.07	0.58 bc ± 0.30	0.26 c ± 0.04	0.43 c ± 0.16	0.31 c ± 0.16	0.46 c ± 0.27	1.35 ab ± 0.60	2.05 a ± 0.25	1.46 a ± 0.28
Oleosins										
ABB01617.1	oleosin low molecular weight isoform; LINUS	0.12 b ± 0.00	0.24 ab ± 0.23	0.03 b ± 0.02	0.23 ab ± 0.15	0.15 ab ± 0.01	0.05 b ± 0.03	0.17 ab ± 0.02	0.51 a ± 0.29	0.11 b ± 0.07
ABB01620.1	oleosin low molecular weight isoform, partial; LINUS	0.17 a ± 0.01	0.29 a ± 0.20	0.03 a ± 0.01	0.30 a ± 0.12	0.17 a ± 0.01	0.06 a ± 0.03	0.26 a ± 0.11	0.38 a ± 0.26	0.12 a ± 0.07
ABB01616.1	oleosin high molecular weight isoform; LINUS	0.98 b ± 0.60	1.11 ab ± 0.46	0.12 c ± 0.03	1.77 a ± 0.11	0.60 bc ± 0.01	0.16 c ± 0.03	0.95 b ± 0.19	1.09 ab ± 0.11	0.41 bc ± 0.10
ABB01624.1	oleosin high molecular weight isoform; LINUS	0.64 abc ± 0.45	0.73 abc ± 0.55	0.14 c ± 0.01	0.18 c ± 0.01	1.11 ab ± 0.16	0.19 c ± 0.02	1.17 ab ± 0.25	1.39 a ± 0.32	0.44 bc ± 0.18
ABB01618.1	oleosin low molecular weight isoform; LINUS	0.14 d ± 0.003	0.18 bc ± 0.01	0.03 f ± 0.01	0.15 cd ± 0.01	0.19 b ± 0.01	0.04 ef ± 0.01	0.17 bc ± 0.01	0.22 a ± 0.01	0.06 e ± 0.004
ABB01619.1	oleosin low molecular weight isoform, partial; LINUS	0.07 ab ± 0.01	0.13 ab ± 0.09	0.01 b ± 0.005	0.14 ab ± 0.05	0.09 ab ± 0.01	0.02 a ± 0.01	0.14 ab ± 0.06	0.19 a ± 0.12	0.06 ab ± 0.04
Defence and stress-related proteins										
WMZ41542.1	chitinase; LINUS	9.38 a ± 0.15	7.63 b ± 0.06	1.28 e ± 0.05	10.39 e ± 0.58	9.94 a ± 0.25	3.14 d ± 0.59	6.18 c ± 0.56	5.25 c ± 0.12	3.21 d ± 0.31
AAW31878.1	chitinase, partial; LINUS	0.06 b ± 0.01	0.02 de ± 0.002	0.004 e ± 0.00004	0.10 a ± 0.0004	0.05 c ± 0.0003	0.01 e ± 0.002	0.07 b ± 0.01	0.03 d ± 0.003	0.01 e ± 0.0003
AMY26620.1	late embryogenesis abundant (lea) group 1-embryo development; LINUS	0.39 a ± 0.03	0.22 bc ± 0.01	0.01 d ± 0.003	0.38 a ± 0.07	0.16 c ± 0.03	0.03 d ± 0.005	0.32 ab ± 0.10	0.21 bc ± 0.02	0.02 d ± 0.01
AVY09180.1	Sequence 3353 from patent US 9878004; UNKNOWN	0.86 a ± 0.01	0.91 a ± 0.05	0.29 c ± 0.01	0.89 a ± 0.08	0.95 a ± 0.04	0.35 bc ± 0.05	0.85 a ± 0.03	0.96 a ± 0.02	0.45 b ± 0.06
CAI0408810.1	unnamed protein product; LINTE	0.46 a ± 0.01	0.22 c ± 0.01	0.04 d ± 0.01	0.35 b ± 0.03	0.20 c ± 0.01	0.04 d ± 0.004	0.38 b ± 0.03	0.18 c ± 0.003	0.06 d ± 0.01
1DWM_A	Chain A, Linum usitatissimum trypsin inhibitor; LINUS	1.28 a ± 0.21	0.60 bc ± 0.07	0.21 cd ± 0.01	1.12 a ± 0.29	0.62 b ± 0.09	0.19 d ± 0.04	0.97 ab ± 0.16	0.69 b ± 0.05	0.21 cd ± 0.05
Other selected important proteins										
AVY09175.1	Sequence 3348 from patent US 9878004; UNKNOWN	1.32 b ± 0.04	1.32 b ± 0.04	0.24 d ± 0.05	1.29 b ± 0.11	1.25 b ± 0.03	0.22 d ± 0.02	1.37 ab ± 0.05	1.48 a ± 0.01	0.55 c ± 0.04
CAI0407981.1	unnamed protein product; LINTE	0.88 c ± 0.02	0.85 c ± 0.01	0.25 e ± 0.08	0.97 bc ± 0.11	0.94 bc ± 0.05	0.29 e ± 0.04	1.11 ab ± 0.07	1.18 a ± 0.12	0.57 d ± 0.05
CAF22093.1	glyceraldehyde 3-phosphate dehydrogenase, partial; LINUS	0.38 ab ± 0.03	0.34 b ± 0.003	0.05 c ± 0.02	0.36 b ± 0.03	0.35 b ± 0.02	0.05 c ± 0.004	0.43 a ± 0.02	0.39 ab ± 0.01	0.10 c ± 0.02
BAL41455.1	fructose-bisphosphate aldolase 1, partial; LINGR	0.33 b ± 0.02	0.33 b ± 0.01	0.06 c ± 0.01	0.33 b ± 0.02	0.32 b ± 0.02	0.08 c ± 0.005	0.39 a ± 0.03	0.40 a ± 0.01	0.10 c ± 0.02
P48417.1	Allene oxide synthase, chloroplastic; LINUS	0.67 b ± 0.03	0.58 cd ± 0.01	0.02 e ± 0.01	0.58 cd ± 0.05	0.56 d ± 0.02	0.02 e ± 0.004	0.66 bc ± 0.04	0.75 a ± 0.03	0.07 e ± 0.01
ABM64783.1	cell wall glycosidase, partial; LINUS	0.04 bc ± 0.003	0.04 cd ± 0.003	0.03 d ± 0.004	0.05 b ± 0.01	0.06 a ± 0.002	0.04 c ± 0.001	0.01 e ± 0.0004	0.01 e ± 0.001	0.01 e ± 0.001
CAI0384355.1	unnamed protein product; LINTE	0.04 a ± 0.002	0.03 bc ± 0.002	0.01 d ± 0.002	0.04 a ± 0.01	0.03 c ± 0.001	0.01 d ± 0.001	0.04 a ± 0.003	0.04 ab ± 0.002	0.01 d ± 0.002
CAI0431450.1	unnamed protein product; LINTE	0.10 ab ± 0.002	0.09 b ± 0.002	0.02 d ± 0.004	0.09 ab ± 0.012	0.09 b ± 0.003	0.03 d ± 0.003	0.11 a ± 0.006	0.11 a ± 0.003	0.06 c ± 0.005
CAI0412769.1	unnamed protein product; LINTE	0.06 ab ± 0.005	0.05 b ± 0.003	0.003 c ± 0.001	0.07 a ± 0.012	0.06 ab ± 0.003	0.004 c ± 0.002	0.05 b ± 0.003	0.06 ab ± 0.004	0.01 c ± 0.003
AVY09204.1	Sequence 3377 from patent US 9878004; UNKNOWN	0.08 b ± 0.004	0.08 b ± 0.003	0.01 d ± 0.002	0.08 b ± 0.009	0.07 b ± 0.002	0.01 d ± 0.001	0.09 a ± 0.003	0.09 ab ± 0.003	0.03 c ± 0.001
CAI0440995.1	unnamed protein product; LINTE	0.12 ab ± 0.007	0.10 b ± 0.006	0.01 c ± 0.003	0.11 ab ± 0.012	0.10 b ± 0.007	0.01 c ± 0.002	0.13 a ± 0.009	0.11 ab ± 0.010	0.02 c ± 0.002
CAI0411675.1	unnamed protein product; LINTE	1.01 ab ± 0.09	1.10 a ± 0.03	0.11 c ± 0.03	0.95 b ± 0.05	1.06 ab ± 0.04	0.10 c ± 0.01	1.09 a ± 0.05	1.12 a ± 0.04	0.22 c ± 0.02
AGN56421.1	non-specific lipid transfer protein 1; LINUS	0.54 a ± 0.03	0.49 ab ± 0.01	0.05 d ± 0.003	0.47 abc ± 0.03	0.43 bc ± 0.04	0.06 d ± 0.01	0.49 abc ± 0.05	0.42 c ± 0.02	0.05 d ± 0.01
CAI0414382.1	unnamed protein product; LINTE	0.26 a ± 0.02	0.25 ab ± 0.07	0.03 c ± 0.002	0.32 a ± 0.06	0.21 ab ± 0.03	0.04 c ± 0.01	0.24 ab ± 0.03	0.30 a ± 0.07	0.13 bc ± 0.04

Note: Different lower-case letters in rows indicate the statistically significant difference at *p* < 0.05 (Tukey HSD test).

## Data Availability

The original contributions presented in the study are included in the article/Appendix A, further inquiries can be directed to the corresponding author.

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
