# Peer review of "Proteomic Profile of Flaxseed (Linum usitatissimum L.) Products as Influenced by Protein Concentration Method and Cultivar"

_foods, 2024, doi:10.3390/foods13091288_

Round 1

Reviewer 1 Report

Comments and Suggestions for Authors

Comments:

Why the particle size of flour affects protein composition. In abstract is not clear

Introduction section: why is considered as superfood if contains toxic compounds such as phytic acid and cyanogenic glycosides.

in section 2.4.. Protein extraction was don in triplicates?, or just the gels? It is most conveniente, in order to see reproducibility, to have 3 extractions.

Abstract:

Lines 21-22... by LC-MS/MS analysis, ..

authors should clarify if analysis was 1DE, 2DE or shotgun

Line 22.. (Agriol, Raciol, and Libra)

Line 56: First, alkaline solubilisation of the.. are precipitated by isoelectric point.

Line 66... Madhusudhan and Singh [17]

Line 115: alkaline solubilisation

Line 121: by reducing the reaction to pH4.5..

which reaction is reduced by decreasing pH?

Line 122: after centrifugation ..

Line 134, Line 148.  hours it was used as h

Line 167.. by 90 min

Footnote tables: you can start it using all long table.

Flaxseed

product

Figure 2. 1DE gel shows that all samples have the same bands, but the protein concentration is different, and as expected the three lines of the same protein is the same profile because is the same sample. Authors should try three different extractions.

But, when the protein concentration is not the same amongst samples, it is hard to conclude that one sample has more or less of one of the proteins. So conclusion must take with precaution.

TAble 3..

as mentioned above, if protein concentration is not good, it is hard to have comparisons amongst samples. In fact, there is any information about protein concentration measured, did the authors used Bradford?, BCA?

as observed in Table 1.. all samples have different protein concentration, then this will cause to have different protein extracted. so it is necessary to quantified the protein in samples used for gel and shotgun analysis and more because it was a label-free shotgun.

Comments on the Quality of English Language

i am not native english speaking, but there are few things to improve

Author Response

For review article

Response to Reviewer 1 Comments

Questions for General Evaluation

Yes

Can be improved

Must be improved

Not applicable

Does the introduction provide sufficient background and include all relevant references?

(x)

( )

( )

( )

Are all the cited references relevant to the research?

(x)

( )

( )

( )

Is the research design appropriate?

(x)

( )

( )

( )

Are the methods adequately described?

( )

(x)

( )

( )

Are the results clearly presented?

(x)

( )

( )

( )

Are the conclusions supported by the results?

(x)

( )

( )

( )

Point-by-point response to Comments and Suggestions for Authors

Why the particle size of flour affects protein composition. In abstract is not clear

The different layers of the flaxseeds or flaxseed cake have inhomogeneous protein content and, as a result of the different hardness of the individual layers of the seed, different particle sizes (with different protein content and protein items) are formed during milling, which can be separated by sieving. In our previous work, Bárta et al. (2021), this topic was addressed, and a sieve mesh size of 250 mm appeared to be the most appropriate. In the Discussion section of the submitted manuscript, the effect of sieving on protein concentration and distribution of protein items is explained. The process cannot be explained in sufficient detail in the abstract due to the limited size of the text.

Introduction section: why is considered as superfood if contains toxic compounds such as phytic acid and cyanogenic glycosides.

Thank you for your comment. You are right that flaxseed contains, in addition to nutritionally and healthfully valuable components, also antinutritional factors (such as phytic acid and cyanogenic glycosides). Other oilseeds also contain these negatively acting substances, e.g. rapeseed glucosinolates. For flaxseed, the total benefits of positive substances outweigh the negative components. We have adopted the term "superfood" for flaxseed from other works: Bueno-Diaz et al. (2022) and Mueed et al. (2022). We have filled in the missing citations of the mentioned works.

in section 2.4.. Protein extraction was don in triplicates?, or just the gels? It is most conveniente, in order to see reproducibility, to have 3 extractions.

Thank you for your comment. Yes, the extraction for SDS-PAGE was performed in triplicate.

Abstract: Lines 21-22... by LC-MS/MS analysis, .. authors should clarify if analysis was 1DE, 2DE or shotgun

The type of protein analysis performed was LC-MS/MS (shotgun proteomics). Information has been added to the abstract.

Line 22.. (Agriol, Raciol, and Libra)

Added a comma after "Raciol" and moved the whole bracket after "cultivars"

Line 56: First, alkaline solubilisation of the.. are precipitated by isoelectric point.

Thank you very much for pointing out the inaccuracy, we have corrected it.

Line 66... Madhusudhan and Singh [17]

Corrected, thank you for the warning.

Line 115: alkaline solubilisation

Corrected, thank you for the warning.

Line 121: by reducing the reaction to pH4.5.. which reaction is reduced by decreasing pH?

The sentence has been modified to: „…by decreasing pH to value 4.5 using 2M HCl.“ We apologize for the misstatement.

Line 122: after centrifugation ..

Modified.

Line 134, Line 148.  hours it was used as h

Corrected, thank you for the warning.

Line 167.. by 90 min

Corrected.

Footnote tables: you can start it using all long table.

Flaxseed product

Revised.

Figure 2. 1DE gel shows that all samples have the same bands, but the protein concentration is different, and as expected the three lines of the same protein is the same profile because is the same sample. Authors should try three different extractions.

But, when the protein concentration is not the same amongst samples, it is hard to conclude that one sample has more or less of one of the proteins. So conclusion must take with precaution.

Thank you for the important advice on commenting on the results. Protein extraction for 1D SDS-PAGE was performed uniformly for all samples: the same volume of extraction buffer was added to the same sample weight. According to BCA analysis, the protein content of the extracts was not the same (depending on sample type and variety), so the protein profiles on the gel do not reflect the same amount of protein. So, you are correct that the conclusions cannot be overstated. This has been mentioned in the manuscript text.

TAble 3..

as mentioned above, if protein concentration is not good, it is hard to have comparisons amongst samples. In fact, there is any information about protein concentration measured, did the authors used Bradford?, BCA?

as observed in Table 1.. all samples have different protein concentration, then this will cause to have different protein extracted. so it is necessary to quantified the protein in samples used for gel and shotgun analysis and more because it was a label-free shotgun.

The evaluation of a concentration of total protein in individual samples (replicates) for proteomic analysis were done in two steps: at protein isolates before FASP processing using calibration on quality control 1D SDS PAGE gels (not related to the SDS-PAGE analysis presented in Figure 2) done in proteomics lab (based on total band optical densities after CBB staining) to ensure processing of the same amount of total protein by FASP and next at peptide level prior final LC-MS/MS analyses using preliminary quality control LC-MS/MS analyses based on calibration curve built up using MEC1 tryptic digests to ensure injection of the same amount of peptides (1 mg) for each individual sample (replicate). In this way we were able to compare relative distribution of proteins in analysed sample types (measured in triplicates).

Comments on the Quality of English Language

i am not native english speaking, but there are few things to improve

Reviewer 2 Report

Comments and Suggestions for Authors

1. Indicate the number of repetitions of the experiment, and add standard deviation to the data in Tables 1 and 3.

2. Indicate the comparison date of the proteome database.

3. Indicate the quantity of proteins identified by each treatment.

4. An appendix should be included by the author to furnish pertinent details on all proteins studied.

5. The author's analysis of differential proteins needs to be strengthened, and the impact of concentration method and cultivar on proteins should be further discussed and analyzed.

6. Line 258, "For detailed evaluation, a set of 33 PGs with high relative allowance in the total process profile (PGs above 0.5%) or being selected to important flaxseed proteins." But there are only 33 proteins in Table 2, where is the important flaxseed protein? More importantly, not only high abundance proteins and important flaxseed proteins, but also proteins with significant differences in abundance under different concentration method and cultivar should be carefully analyzed and discussed.

Author Response

For review article

Response to Reviewer 2 Comments

Questions for General Evaluation

Yes

Can be improved

Must be improved

Not applicable

Does the introduction provide sufficient background and include all relevant references?

( )

(x)

( )

( )

Are all the cited references relevant to the research?

(x)

( )

( )

( )

Is the research design appropriate?

(x)

( )

( )

( )

Are the methods adequately described?

(x)

( )

( )

( )

Are the results clearly presented?

( )

( )

(x)

( )

Are the conclusions supported by the results?

( )

(x)

( )

( )

Point-by-point response to Comments and Suggestions for Authors

  1. Indicate the number of repetitions of the experiment, and add standard deviation to the data in Tables 1 and 3.

All sample analyses reported in the present work were performed in triplicate. Standard deviations to the presented values have been added to Tables 1 and 3.

  1. Indicate the comparison date of the proteome database.

The version of database which was used for database searching is already mentioned in method section in detail: „whole Linum genus database (https://www.ncbi.nlm.nih.gov/ipg/?term=txid4005[Organism:exp]; version 2023-09-30, number of protein sequences: 125,854).“

We chose commonly used way of database description in proteomic studies defining the database version by the release date (2023-09-30) and we are ready to provide the applied database version on request.

  1. Indicate the quantity of proteins identified by each treatment.

The amount of protein contained in the different variants of flax products - seed cake, fine flour, protein concentrate (that result from sieving or from alkaline solubilisation/isoelectric precipitation process) - in combination with the evaluated cultivars is given in Table 1 in the form of Nx6.25. In the original version of the manuscript, an error was made by copying the data into the manuscript template (as also pointed out by other reviewers); in the revised version, the data are given correctly. We apologise for this error.

  1. An appendix should be included by the author to furnish pertinent details on all proteins studied.

The table containing list of identified proteins groups, protein abundancy and other details was attached as Supplementary Table 1.

  1. The author's analysis of differential proteins needs to be strengthened, and the impact of concentration method and cultivar on proteins should be further discussed and analyzed.

We have tried to reinforce these recommendations in the discussion section of the manuscript.

  1. Line 258, "For detailed evaluation, a set of 33 PGs with high relative allowance in the total process profile (PGs above 0.5%) or being selected to important flaxseed proteins." But there are only 33 proteins in Table 2, where is the important flaxseed protein? More importantly, not only high abundance proteins and important flaxseed proteins, but also proteins with significant differences in abundance under different concentration method and cultivar should be carefully analyzed and discussed.

We agree with your view that the importance of a protein cannot be assessed solely on the basis of relative abundance. Table 2 lists both the major proteins (11S globulins, 2S albumins or conlinins, oleosins) and other metabolically important proteins, which, for example, had significantly different abundances in different flaxseed products. For completeness of the data, at your suggestion, we have supplemented the manuscript with a supplementary table listing all the proteins analyzed.

Reviewer 3 Report

Comments and Suggestions for Authors

The ms. “Proteomic profile of flaxseed (Linum usitatissimum L.) products as influenced of protein concentration method and cultivar” (Ms. Ref. No. foods-2933299-v1) presents original results on the flaxseed protein characterization regarding the main protein fractions from seed cake (FC), fine flaxseed flour (FF) and protein concentrate (PC). Three flaxseed cultivars were investigated (Agriol, Raciol and Libra).

 There is a lot of work involved.

The topic falls within the aims and scopes of the Foods journal.

However, there are important issues that need to be addressed to improve the presentation of the results.

Major issues:

1.     In the Abstract (and the manuscript): The abbreviation FC for the seed cake is not intuitive in English. I suppose it may be intuitive in the authors’ mother tongue, however I suggest changing FC to SC. If you agree with this suggestion, please make the necessary changes throughout the ms. (including tables, figures, etc.).

2. In the Introduction: Please better point out the elements of originality/novelty of your work with respect to the current literature landscape. For example, if the methodology used in this study to concentrate the proteins is preventing protein denaturation compared to other studies, it should be pointed out.

3.     Also in the Introduction: it is worth pointing out the amino acids profiles of the flaxseed proteins, which are abundant in Glu, Asp and Arg (see https://doi.org/10.3390/foods11131820). At the same time, if other essential amino acids are well represented, they should be mentioned. Please revise.

4.     Lines 44-45: Since n-3 fatty acids are more biologically active than n-6 and given n-3 ALA is the main constituent of the flaxseed oil, the text should be re-phrased as to point out ALA first and then n-6. Also, for linolenic and linoleic acids I suggest giving their lipid numbers (C18:3 and C18:2, respectively).

5.     The specific contribution of the linseed flour to food products should be mentioned in the Introduction (or in the Discussion section), see, for example https://doi.org/10.3390/foods11071022 and https://doi.org/10.3390/foods9101383.

6.     Lines 47-48: Regarding the drying properties: I suggest mentioning the high iodine value (up to 205.2 gI2/100 g oil, according to Anastasiu, 2016, Oil productivity of seven Romanian linseed varieties as affected by weather conditions), as there are only very few species with higher unsaturated oil (such as Lalllemantia iberica, see Komartin et al., Optimization of oil extraction from Lallemantia iberica seeds using ultrasound-assisted extraction).

7.     The motivation of the study (that there is few knowledge on the characterization of the flaxseed proteins) is weak. I suggest improving the motivation of selecting flaxseed as a source of proteins against other species. Flaxseed/linseed are mainly cultivated for their highly un-saturated oil. So is – for example - Lallemantia iberica (with even higher unsaturation degree, see above). However, linseed cake can provide important amounts of quality proteins, while Lallemantia may contain toxic minor constituents, which may extract along with the proteins, thus making it unsuitable as a source of proteins.

8.     In the M&M section: Please mention the agronomic conditions and practices for the three linseed crops, including crop year.

9.     Regarding Table 1: It is striking that the data are identical for the three cultivars, for all the products (i.e. WS, FC, FF and PC). I suspect there is a matter of involuntary copy-paste issue, as the corresponding text in lines 203-208 reads about “noticeable differences” among cultivars. Please check and update Table 1.

10.  Also regarding Table 1: I suggest using lower-case letters to showcase the statistical differences among the flaxseed products (i.e. WS, FC, FF and PC) within each cultivar and Capital letters to point out the differences among the three cultivars (within each product). Of course, do not forget to update the table footnote!

Minor issues:

1.     Title: I suggest rephrasing as: “Proteomic profile of flaxseed (Linum usitatissimum L.) products as influenced by protein concentration method and cultivar”.

2.     Please conform to the standards and write all the Latin names (scientific names of the species) with Capital first letter and Italics.

3. In general, English is ok. However, there are a few syntax/grammar/typos which I suppose will be fixed in later manuscript processing stages.

Given the completed score sheet and the comments above, after careful evaluation, the ms. “Proteomic profile of flaxseed (Linum usitatissimum L.) products as influenced of protein concentration method and cultivar” (Ms. Ref. No. foods-2933299-v1) needs Major Revision according to comments.

Comments on the Quality of English Language

In general, English is ok. However, there are a few syntax/grammar/typos which I suppose will be fixed in later manuscript processing stages.

Author Response

For review article

Response to Reviewer 3 Comments

Questions for General Evaluation

Yes

Can be improved

Must be improved

Not applicable

Does the introduction provide sufficient background and include all relevant references?

( )

( )

(x)

( )

Are all the cited references relevant to the research?

(x)

( )

( )

( )

Is the research design appropriate?

( )

(x)

( )

( )

Are the methods adequately described?

( )

(x)

( )

( )

Are the results clearly presented?

( )

(x)

( )

( )

Are the conclusions supported by the results?

(x)

( )

( )

( )

Point-by-point response to Comments and Suggestions for Authors

The ms. “Proteomic profile of flaxseed (Linum usitatissimum L.) products as influenced of protein concentration method and cultivar” (Ms. Ref. No. foods-2933299-v1) presents original results on the flaxseed protein characterization regarding the main protein fractions from seed cake (FC), fine flaxseed flour (FF) and protein concentrate (PC). Three flaxseed cultivars were investigated (Agriol, Raciol and Libra).

 There is a lot of work involved.

The topic falls within the aims and scopes of the Foods journal.

However, there are important issues that need to be addressed to improve the presentation of the results.

Major issues:

  1. In the Abstract (and the manuscript): The abbreviation FC for the seed cake is not intuitive in English. I suppose it may be intuitive in the authors’ mother tongue, however I suggest changing FC to SC. If you agree with this suggestion, please make the necessary changes throughout the ms. (including tables, figures, etc.).

Thank you for the important comment on the abbreviations used, we have changed the abbreviation FC to the recommended abbreviation SC in the manuscript.

  1. In the Introduction: Please better point out the elements of originality/novelty of your work with respect to the current literature landscape. For example, if the methodology used in this study to concentrate the proteins is preventing protein denaturation compared to other studies, it should be pointed out.

Thank you for the important reminder. In the introduction section, the revised version of the manuscript includes the information that the evaluation of protein profiles of flaxseeds and their derived products (flour or protein concentrate) using proteomic approaches has not received sufficient attention or relevant information is not available. Thus, the information reported on flaxseed proteins in our work is new.

  1. Also in the Introduction: it is worth pointing out the amino acids profiles of the flaxseed proteins, which are abundant in Glu, Asp and Arg (see https://doi.org/10.3390/foods11131820). At the same time, if other essential amino acids are well represented, they should be mentioned. Please revise.

Additional text on amino acids has been added to the introduction section, as per your suggestion. Thank you for the idea.

  1. Lines 44-45: Since n-3 fatty acids are more biologically active than n-6 and given n-3 ALA is the main constituent of the flaxseed oil, the text should be re-phrased as to point out ALA first and then n-6. Also, for linolenic and linoleic acids I suggest giving their lipid numbers (C18:3 and C18:2, respectively).

The text has been modified as proposed.

  1. The specific contribution of the linseed flour to food products should be mentioned in the Introduction (or in the Discussion section), see, for example https://doi.org/10.3390/foods11071022 and https://doi.org/10.3390/foods9101383.

Additional text expanding on the application of flaxseed products has been added to the introduction section, and suggested articles have been cited.

  1. Lines 47-48: Regarding the drying properties: I suggest mentioning the high iodine value (up to 205.2 gI2/100 g oil, according to Anastasiu, 2016, Oil productivity of seven Romanian linseed varieties as affected by weather conditions), as there are only very few species with higher unsaturated oil (such as Lalllemantia iberica, see Komartin et al., Optimization of oil extraction from Lallemantia ibericaseeds using ultrasound-assisted extraction).

Additional information on drying oils has been added to the introduction section, and suggested articles have been cited.

  1. The motivation of the study (that there is few knowledge on the characterization of the flaxseed proteins) is weak. I suggest improving the motivation of selecting flaxseed as a source of proteins against other species. Flaxseed/linseed are mainly cultivated for their highly un-saturated oil. So is – for example - Lallemantia iberica(with even higher unsaturation degree, see above). However, linseed cake can provide important amounts of quality proteins, while Lallemantia may contain toxic minor constituents, which may extract along with the proteins, thus making it unsuitable as a source of proteins.

We agree with you that flax is cultivated for the specific quality of its oil (a drying oil with a high ALA content) and that, for example, the oil seeds of Lallemantia iberica plant has an oil with similar properties. However, flax is also grown for food purposes only – seed, nutritional oil, flour – and so information on the protein profile of flaxseed and its derivatives is important in its own right. However, in the text of the introduction section we have tried to better explain the novelty and the need for more detailed data on the protein profiles of flaxseeds.

  1. In the M&M section: Please mention the agronomic conditions and practices for the three linseed crops, including crop year.

Information on flax cultivation, harvesting and storage of seeds has been inserted in the M&M section.

  1. Regarding Table 1: It is striking that the data are identical for the three cultivars, for all the products (i.e. WS, FC, FF and PC). I suspect there is a matter of involuntary copy-paste issue, as the corresponding text in lines 203-208 reads about “noticeable differences” among cultivars. Please check and update Table 1.

You are right, there was a mistake with copying the table data into the template of the manuscript. The table is now corrected. We apologize to the reviewers and the editors for the mistake.

  1. Also regarding Table 1: I suggest using lower-case letters to showcase the statistical differences among the flaxseed products (i.e. WS, FC, FF and PC) within each cultivar and Capital letters to point out the differences among the three cultivars (within each product). Of course, do not forget to update the table footnote!

 Table 1 has been revised including the proposed statistical evaluation.

Minor issues:

  1. Title: I suggest rephrasing as: “Proteomic profile of flaxseed (Linum usitatissimumL.) products as influenced by protein concentration method and cultivar”.

The title of the work has been modified to the proposed form.

  1. Please conform to the standards and write all the Latin names (scientific names of the species) with Capital first letter and Italics.

The text of the manuscript has been revised and the Latin names of plant species are now in italics and with Capital first letter.

  1. In general, English is ok. However, there are a few syntax/grammar/typos which I suppose will be fixed in later manuscript processing stages.

We agree with your opinion.

Given the completed score sheet and the comments above, after careful evaluation, the ms. “Proteomic profile of flaxseed (Linum usitatissimum L.) products as influenced of protein concentration method and cultivar” (Ms. Ref. No. foods-2933299-v1) needs Major Revision according to comments.

Comments on the Quality of English Language

In general, English is ok. However, there are a few syntax/grammar/typos which I suppose will be fixed in later manuscript processing stages.

Reviewer 4 Report

Comments and Suggestions for Authors

Aim, novelty and significance.

The aim of the study to contribute to the extension of our knowledge about flaxseed protein composition in three model cultivars and at the level of flaxseed cake, fine flaxseed flour and protein concentrate.

The aim of the study is quite significant and respond to the existing need to extend our knowledge about this protein since the information on the flaxseed proteome is scarce.

However, the article did not explain the hypothesis behind the potential difference between the three products used to conduct the analysis (FC, FF and PC).

For the FF product, it is not clear why the work used only one mesh size of 250 µm, if the authors are anticipating the size will be an influencing factor, why did not they test different sizes?

The introduction should clarify the importance of the different protein subunits of flaxseed proteins particularly 11S, so that people interested in the protein may be motivated to further follow and extract this protein from this unusual source. There is a plethora of research works on this protein subunit and its biological activity. Referring to these works will further enhance the significance of the current work..

Title

The title should be grammatically corrected into (Proteomic profile of flaxseed (Linum usitatissimum L.) products as influenced by protein concentration method and cultivar)

The abbreviation FC is used to represent Flaxseed cake as it is shown in the material and metods L112 [flaxseed cake (FC)]. However in the abstract (line 20), it is introduced as (seed cake (FC)). This is confusing. The signification of the abbreviation should be strictly the same allover the article.

Materials and Methods

It is mentioned that Proteins from flaxseed product samples were extracted in SDT buffer (4% SDS, 0.1M DTT, 0.1M Tris/HCl, pH 7.6). Then the supernant was used for tryptic analysis.

Did the author make sure that the intended proteins were totally soluble at the basic pH. It is known that 11S protein one of the major components of these protein products may be hardly soluble at this neutral pH. Did the author calculate the isoelectric points of the different protein fractions to avoid the absence or reduced levels of any fractions in the final analysis.

Since the article is to elucidate the flaxseed protein composition, it will be meaningful the isolate the different protein fractions, particularly 11S and 2S, and characterize chemically them, for example by SDS-PAGE, UREA-PAGE and isoelectric points.

Results and Discussion

In this section the results should be discussed compared to the initial hypothesis of the article. If for example the cultivar factor was not influencing the proteomic profile of flaxseed products, the article should provide explanations and reasons and also modify the conclusion accordingly.

L251, it is mentioned (discussed more in Chapter 3.3). The use of the word (chapter) is not correct in this context (chapter means a part of a book). The words (section or entry) can be used instead.

Conclusions

L468, change into (descriptions) and (the identified)  

L470, it is mentioned [The conditions combining alkaline extraction (at pH 4.5)]. Please verify and correct.

The article should clearly indicate which cultivar and which protein products were the best to obtain the protein fractions of interest e.g. 11S and 2S. The conclusion should also refer to the initial hypothesis of the work concluding if the work done approved or disapproved the hypothesis.

Comments on the Quality of English Language

Moderate editing is beneficial

Author Response

For review article

Response to Reviewer 4 Comments

Questions for General Evaluation

Yes

Can be improved

Must be improved

Not applicable

Does the introduction provide sufficient background and include all relevant references?

(x)

( )

( )

( )

Are all the cited references relevant to the research?

(x)

( )

( )

( )

Is the research design appropriate?

( )

(x)

( )

( )

Are the methods adequately described?

( )

(x)

( )

( )

Are the results clearly presented?

( )

(x)

( )

( )

Are the conclusions supported by the results?

( )

(x)

( )

( )

Point-by-point response to Comments and Suggestions for Authors

Aim, novelty and significance.

The aim of the study to contribute to the extension of our knowledge about flaxseed protein composition in three model cultivars and at the level of flaxseed cake, fine flaxseed flour and protein concentrate.

The aim of the study is quite significant and respond to the existing need to extend our knowledge about this protein since the information on the flaxseed proteome is scarce.

However, the article did not explain the hypothesis behind the potential difference between the three products used to conduct the analysis (FC, FF and PC).

In the revised version of the manuscript (in the Introduction, Results and Discussion sections as well as in Figure 1), the differences and importance of the three flaxseed products evaluated (cake, flour and protein concentrate) were better explained. The product names and their acronyms (SC - seed cake, FF - fine flour, PC - protein concentrate) were modified based on the request of other opponents and the academic editor.

For the FF product, it is not clear why the work used only one mesh size of 250 µm, if the authors are anticipating the size will be an influencing factor, why did not they test different sizes?

The sieve mesh size 250 mm was chosen based on our previous experience, when we tried different sieve mesh sizes in the range 180-750 mm and the mesh size 250 mm was able to provide the best results in terms of concentration effect and practical implementation, because the sieve with a lower mesh size was already heavily clogged, which complicated the actual fractionation process. This experience led to the use of the mesh size sieve also in our previous work by Bárta et al. (2021).

The introduction should clarify the importance of the different protein subunits of flaxseed proteins particularly 11S, so that people interested in the protein may be motivated to further follow and extract this protein from this unusual source. There is a plethora of research works on this protein subunit and its biological activity. Referring to these works will further enhance the significance of the current work..

In the text of the Introduction section, references have been added to recent research papers dealing with the application of flaxseed proteins (especially 11S globulins and 2S albumins) in food applications, their impact on human nutrition and medical applications.

Title

The title should be grammatically corrected into (Proteomic profile of flaxseed (Linum usitatissimum L.) products as influence by protein concentration method and cultivar)

Corrected, thank you for the warning.

The abbreviation FC is used to represent Flaxseed cake as it is shown in the material and metods L112 [flaxseed cake (FC)]. However in the abstract (line 20), it is introduced as (seed cake (FC)). This is confusing. The signification of the abbreviation should be strictly the same allover the article.

The flaxseed cake product is now designated as seed cake with the acronym SC and is thus used throughout the manuscript. Thank you for the notice.

Materials and Methods

It is mentioned that Proteins from flaxseed product samples were extracted in SDT buffer (4% SDS, 0.1M DTT, 0.1M Tris/HCl, pH 7.6). Then the supernant was used for tryptic analysis.

Did the author make sure that the intended proteins were totally soluble at the basic pH. It is known that 11S protein one of the major components of these protein products may be hardly soluble at this neutral pH. Did the author calculate the isoelectric points of the different protein fractions to avoid the absence or reduced levels of any fractions in the final analysis.

In our study, we used SDT buffer which contains 4% sodium dodecyl sulphate (SDS), a strong and widely used detergent which ensures efficient extraction of proteins from various matrices even in neutral pH. We use this buffer in combination with FASP successfully for a long time and for wide range of matrices including meat products, plant tissues, grains etc.

The details about applied sample prep method and SDS might be find in Jacek R. Wisniewski, Analytica Chimica Acta 1090 (2019), 23-30.

Since the article is to elucidate the flaxseed protein composition, it will be meaningful the isolate the different protein fractions, particularly 11S and 2S, and characterize chemically them, for example by SDS-PAGE, UREA-PAGE and isoelectric points.

You are right that 11S and 2S proteins are the major proteins of flaxseed and could be analyzed by other methods. However, the article is thematically focused on the assessment of protein profiles using LC-MS/MS techniques followed by estimation of the relative abundance of the identified protein groups. In a follow-up work, we plan a detailed evaluation of 11S globulins and 2S albumins and the proposed methods will be used for detailed biochemical characterization of these proteins.

Results and Discussion

In this section the results should be discussed compared to the initial hypothesis of the article. If for example the cultivar factor was not influencing the proteomic profile of flaxseed products, the article should provide explanations and reasons and also modify the conclusion accordingly.

L251, it is mentioned (discussed more in Chapter 3.3). The use of the word (chapter) is not correct in this context (chapter means a part of a book). The words (section or entry) can be used instead.

Thank you for your recommendation, the text of the Results and Discussion section has been expanded to include the discussion section in line with the newly revised objectives of the article. The findings have been incorporated into the text of the Conclusion section. The recommended change of the term chapter to the term section has been reflected in the text of the manuscript.

Conclusions

L468, change into (descriptions) and (the identified)  

It has been performed.

L470, it is mentioned [The conditions combining alkaline extraction (at pH 4.5)]. Please verify and correct.

Thank you for pointing out the error, it should have been pH 8.5 instead of pH 4.5. Corrected.

The article should clearly indicate which cultivar and which protein products were the best to obtain the protein fractions of interest e.g. 11S and 2S. The conclusion should also refer to the initial hypothesis of the work concluding if the work done approved or disapproved the hypothesis.

 Thank you for your comment, which we have taken into account in the Conclusion section.

Comments on the Quality of English Language

Moderate editing is beneficial

Round 2

Reviewer 2 Report

Comments and Suggestions for Authors

This manuscript has been sufficiently improved.

Author Response

For review article

Response to Reviewer 2 Comments (Round 2)

Questions for General Evaluation

Yes

Can be improved

Must be improved

Not applicable

Does the introduction provide sufficient background and include all relevant references?

(x)

( )

( )

( )

Are all the cited references relevant to the research?

(x)

( )

( )

( )

Is the research design appropriate?

(x)

( )

( )

( )

Are the methods adequately described?

(x)

( )

( )

( )

Are the results clearly presented?

(x)

( )

( )

( )

Are the conclusions supported by the results?

(x)

( )

( )

( )

Point-by-point response to Comments and Suggestions for Authors

This manuscript has been sufficiently improved.

Thank you for your review.

Reviewer 4 Report

Comments and Suggestions for Authors

L50, change (which indicates the unsaturation of the oils,) t0 (which indicates the unsaturation level of the oils,).

L324-325, change into (However, this suggestion can be taken with some reserve because the absolute amount of protein in the samples loaded on the electrophoretic gel differed.)

It will be useful for the interested readers to comprehend the previous experience of the authors behind using only one mesh size of 250 µm as they explained in their responses. This information should be incorporated some relevant location in the text.

Comments on the Quality of English Language

Only slight ordinary editing may be required

Author Response

For review article

Response to Reviewer 4 Comments (Round 2)

Questions for General Evaluation

Yes

Can be improved

Must be improved

Not applicable

Does the introduction provide sufficient background and include all relevant references?

(x)

( )

( )

( )

Are all the cited references relevant to the research?

(x)

( )

( )

( )

Is the research design appropriate?

(x)

( )

( )

( )

Are the methods adequately described?

(x)

( )

( )

( )

Are the results clearly presented?

(x)

( )

( )

( )

Are the conclusions supported by the results?

(x)

( )

( )

( )

Point-by-point response to Comments and Suggestions for Authors

L50, change (which indicates the unsaturation of the oils,) t0 (which indicates the unsaturation level of the oils,).

Modified as proposed.

L324-325, change into (However, this suggestion can be taken with some reserve because the absolute amount of protein in the samples loaded on the electrophoretic gel differed.)

Modified as proposed.

It will be useful for the interested readers to comprehend the previous experience of the authors behind using only one mesh size of 250 µm as they explained in their responses. This information should be incorporated some relevant location in the text.

Thank you for the recommendation. The text explaining sieve selection has been added to section 3.3. (L436-439).
